

# The role of objectively recorded smartphone usage and personality traits in sleep quality

Aftab Alam, Sameha Alshakhsi, Dena Al-Thani and Raian Ali

College of Science and Engineering, Hamad Bin Khalifa University, Doha, Qatar

## ABSTRACT

**Purpose**. The proliferation of smartphones, accompanied by internet facilities, has contributed to a decrease in sleep quality over the last decades. It has been revealed that excessive internet usage impacts the physical and mental health of smartphone users, while personality traits (PT) could play a role in developing internet addictions and preventing their negative effects. The objective of the present study is to assess the role of PT and smartphone usage in sleep quality.

**Method**. The sample comprised 269 participants, 55% females, within the age range of 15–64 years. We objectively collected one-week smartphone apps usage data from the participants. They also responded to demographics and the PT (BFI-10) questionnaires. The usage data of smartphone apps were processed to calculate smartphone usage amounts and sleep variables, including sleep duration, sleep distraction, sleeping time, and wake-up time. The data were analyzed using the correlation coefficient and regression analyses.

**Results**. The results indicated that more smartphone usage was associated with reduced sleep duration, increased sleep distraction, and later bedtime. Furthermore, smartphone users with the conscientiousness trait had a longer sleep duration, earlier sleeping time, less sleep distraction, and earlier wakeablity. Sleep distraction was positively associated with openness. Extraversion and neuroticism were found to be positive predictors of early wakeablity. Neuroticism had a negative association with early wakeablity. Finally, the implications of the study have been discussed.

**Conclusion**. Our study's usage of data that was acquired objectively has strong methodological qualities. The present study is the first to contribute to the literature on the role of PT and objectively measured smartphone usage in the prediction of sleep quality. We found that smartphone use and sleep variables are associated with PT. Other scholars can use our dataset for benchmarking and future comparisons.

Corresponding authors
Aftab Alam, aftabdir@gmail.com
Raian Ali, raali2@hbku.edu.qa

## INTRODUCTION

The smartphone's recent developments, as well as its global popularity, have changed the landscape of communication and information sharing, reshaping users' interests, values, and demands. Smartphones have already begun to substantially influence individuals, markets, corporations, and society (*Sarwar & Soomro, 2013*). Researchers are investigating

both the positive and negative consequences of smartphones for consumers, service providers, and society as a whole (*Gonçalves et al., 2020*).

Smartphones played a significant role for individuals of all sectors, especially during the lockdown when the internet allowed people to work from home and enhanced the flow of business tasks and responsibilities (*Iyengar et al., 2020*; *WHO, 2020b*). Similarly, easy and constant access to smartphones connected to the internet aided students in their educational and professional pursuits (*Iyengar et al., 2020*). Furthermore, the smartphone enables young people to practice social skills and avoid the social anxiety, stress, and peer influences associated with face-to-face interactions (*Heitner, 2002*; *McKenna, Green & Gleason, 2002*).

Smartphone use, on the other hand, may have negative consequences on users' well-being. Low self-esteem, poor sleep quality, mood disorders, and poor daily work performance are among several personality and psychiatric issues linked to excessive smartphone usage (*Tripathi, 2017*). Socially anxious and depressed people have been reported to be more likely to use the internet excessively as a coping technique to escape sad thoughts, negative emotions, and in some cases, drug use (*Armstrong, Phillips & Saling, 2000*; *Lee et al., 2013*). The chances are high of getting addicted to the smartphone and technology in general by spending more time online, disrupting their physical and psychological functioning (*Cash et al., 2012*). Technology addiction has been defined as "non-chemical, behavioral addictions that involve human-machine interactions" (*Griffiths, 1996*). Gaming Disorder, which includes both online and offline gaming, has been added to the World Health Organization's International Classification of Diseases (ICD-11) as an official diagnosis (*Darvesh et al., 2020*; *WHO, 2020a*).

In literature, smartphone addiction has been defined as the overuse of smartphones to the extent that it disturbs the users' daily life priorities, including sleep (*Demirci et al., 2014*). Sleep is vital for health, everyday functioning, and performance, and it is "the most common, important, and potentially remediable health risk in teens" (*Sheldon, 2015*). Sleep deprivation has been connected to concerns with both physical and mental health, including poor immune response (*Besedovsky, Lange & Born, 2012*), mental illness (*Marino et al., 2018*), low fertility (*Kloss et al., 2015*), impaired cognition (*Lim & Dinges, 2010*), hypertension (*Palagini et al., 2013*), and hyperglycemia (*Reutrakul et al., 2018*). People will increase their internet usage and are more likely to disrupt their sleep-wake cycle, which could result in an increased rate of insomnia (*Jenaro et al., 2007*), fatigue during the day, and poor sleeping patterns (*Punamäki et al., 2007*). The delay in sleeping time and reduced sleep duration result in excessive fatigue, impaired academic or professional performance, and a weakened immune system (*Garbarino et al., 2021*; *Maheshwari & Shaukat, 2019*).

Many aspects of human behavior, including music listening and the rate at which people utilize technology, particularly the internet and digital devices, are thought to be influenced by PT (*Amichai-Hamburger, 2002*; *Meston, Trapnell & Gorzalka, 1996*). According to *Gunduz, Eksioglu & Tarhan (2017)*, increased neuroticism is linked to excessive internet use, whereas conscientiousness is linked to decreased internet use. People with negative PT have trouble falling asleep, while individuals with positive PT report having healthy sleep habits (*Kim et al., 2015*). Poor sleep quality increases daytime sleepiness, and complaints

about poor sleep are typically associated with people who have high levels of neuroticism (*Gray & Watson, 2002*).

Even though researchers have investigated the negative impacts of excessive internet use, studies in the literature have relied on self-reported smartphone usage and sleep quality, which have significant methodological limitations. Self-reported technology usage does not match actual technology usage (*Boase & Ling, 2013*; *Kobayashi & Boase, 2012*). In this study, we examine the role of smartphone usage and personality traits (PT) in the prediction of sleep quality, where both the usage and the sleep parameters are objectively collected. In this work, we aim to study the following research questions:

- Do PT and smartphone usage relate to sleep duration while considering age and gender?
- Does sleep distraction caused by smartphone usage while sleeping relate to PTs and smartphone usage?
- Is sleeping time influenced by the usage of smartphones and PT, age, and gender?
- Do PT and smartphone usage have relationships with wakeup time while considering age, and gender?

## RELATED WORK

Many studies have shown that getting enough sleep is significant for our health. Sleep is a natural physical and mental resting state in which a person becomes passive and unconscious of their surroundings (*Lou et al., 2004*; *Posner et al., 2007*). It is distinguished by altered consciousness, reduced sensory activity, and inhibition of nearly all voluntary muscles. During sleep, the capacity to react to stimuli is reduced. Certain relays from specific brain circuits or brain activity govern sleep patterns, which are controlled by the circadian clock and sleep-wake homeostasis. The brain has a 24-hour internal biological clock that is influenced by the individual's physical and mental health (*Walker et al., 2020*). Consistent bedtimes, sleep duration, distraction less sleep, and wake-up times encourage healthy sleep-wake cycles by promoting circadian rhythms. Cycling through all phases of sleep for a suitable period is necessary for total rest. Though research findings on the duration of adequate sleep hours for adolescents vary (*Honkus, 2003*; *Knutson et al., 2007*), most studies show that approximately seven hours of sleep are sufficient for tissue restoration, the release of growth hormones, and efficient healthy psychological functioning during adolescence (*Chaput, Dutil & Sampasa-Kanyinga, 2018*).

Researchers have revealed that students during the lockdown, especially during infection with COVID-19, used the internet more at nighttime, had disturbed sleep, had higher levels of internet addiction, and had changed their sleep patterns. (*Nayak et al., 2021*; *Tahir et al., 2021*). Wake-up delay, sleeping delay, any-time sleepability, and wakeability are examples of sleep quantity and quality disruptions related to such interruptions during sleep time (*Cherepanova & Putilov, 1991*; *Danilenko et al., 2004*; *Melnikov et al., 1999*). Although personality is a significant predictor of good sleep (*Duggan et al., 2014*), less is known regarding the relationship between personality and sleep quality of smartphone users. Therefore, this study looked at the influence of PT on people's sleeping quality while

considering the daily average smartphone usage that is recorded objectively through a dedicated app.

An individual's personality is a collection of structured and relatively long-lasting psychological qualities and mechanisms that impact their interactions with and adaptations to intrapsychic, physical, and social contexts (*Allport, 1937*). Personality is a reasonably constant pattern of thoughts, feelings, social adaptations, and actions that manifests itself in various contexts (*Feist, 1994*). Personality influences how people move and react to their surroundings. PT is thought to influence a wide range of human behaviors, including music listening and the pace at which individuals use technology, particularly the internet and digital devices (*Amichai-Hamburger, 2002*; *Meston, Trapnell & Gorzalka, 1996*). A study by *Gunduz, Eksioglu & Tarhan (2017)* found that an increase in the neuroticism trait is associated with excessive internet use, whereas the conscientiousness trait is associated with decreased internet use. Several sleep studies examined the importance of personality qualities in people's sleep habits (*Hintsanen et al., 2014*; *Yañez et al., 2020*). There is considerable research on the relationship between PT and sleep patterns among middle-aged and older citizens, which demonstrated poor sleep quality among adults low on extraversion, conscientiousness, and agreeableness (*Stephan et al., 2018*). Individuals with negative PT are poor sleepers, whereas those with positive PT report being good sleepers with appropriate sleep patterns (*Kim et al., 2015*). Research on how personality affects sleep has revealed contradicting results. The study by *Soehner, Kennedy & Monk (2007)* found no link between PT and sleep duration. Another study by *Gamaldo et al. (2020)* claimed that the extroversion trait contributes to improved sleep more than the introversion trait. People with high neurotic personalities, on the other hand, are frequently linked to poor sleep quality, increased daytime sleepiness, and complaints about poor sleep (*Gray & Watson, 2002*).

Existing research work overlooked the prediction of sleep variables and sleep/wake patterns using objectively measured smartphone usage and PT. Thus, this research aims to recognize the relationship between PT and objectively recorded smartphone usage with sleep variables while considering demographic information like age and gender.

# RESEARCH METHODS

## Study design

An overview of our study is shown in Fig. 1. The participants were first allowed to install a dedicated smartphone app for monitoring the time they spend on the smartphone and the apps used. Upon installing the app, the participants were explicitly asked to consent to the data collection and the use of their data for research on digital well-being, anonymously. After app installation, the participants completed the 10-item Big Five Inventory (BFI-10) (*Rammstedt & John, 2007*) along with demographic information; education, gender, and age. In return, the participants were offered a premium version of the app as a token of appreciation for their participation. Upon consenting to take part in the study and completing the survey, the collection of smartphone usage data started. Upon consenting to take part in the study and completing the survey, the collection of smartphone usage data

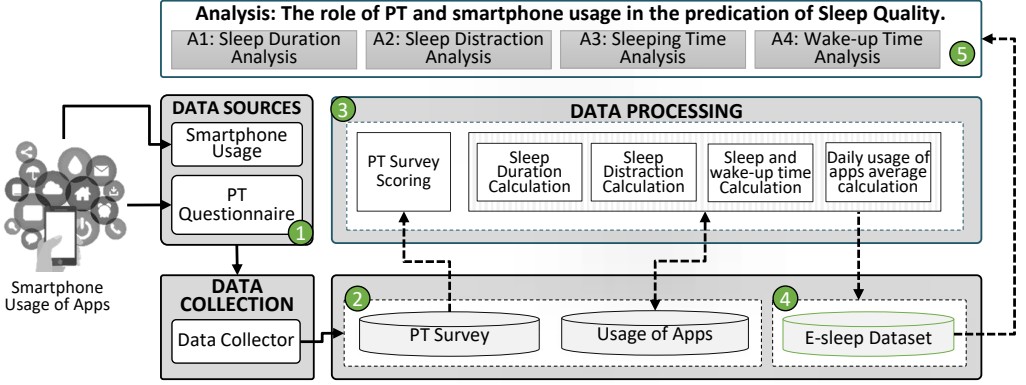

**Figure 1** An abstract view of our study.

started. Then, the acquired data were processed to calculate the BFI-10 score and e-sleep variables. Based on the sleep-wake cycle, the sleep variables have been extracted from the usage data of the apps. Thus, we use the term "e-sleep" in this study. E-sleep represents the passive time of smartphones during sleeping hours. There are chances that the smartphone user might be awake but not using the smartphone during sleeping hours. The purpose of this study is to determine how much time users spend away from their smartphones and how it causes distraction during sleeping hours. All the extracted information was then stored in the e-sleep dataset. Finally, analysis was performed on the e-sleep dataset ($N = 269$) using JASP version 0.16.0 (*Goss-Sampson, 2019*) for multivariate linear regression and logistic regression analysis and SPSS version 28.1.1.1 (*Hinton, McMurray & Brownlow, 2014*) for multinomial logistic regression analysis.

## MEASURES

### Personality assessment

To assess the PT, we used Big Five Inventory-10 (BFI-10) (*Rammstedt & John, 2007*). This 10-item self-report questionnaire assesses extraversion, agreeableness, conscientiousness, neuroticism, and openness to experience, the five personality traits. Participants were asked to rate how accurate statements were in describing their personalities. The 10 items are assessed on a 5-point Likert scale (1 = Strongly disagree to 5 = Strongly agree), with each dimension of personality corresponding to two items and, thus, having a value between 2 and 10. A higher score denotes a greater degree of a particular personality trait. The BFI-10 is reliable and valid across a wide range of sample groups (*Rammstedt & John, 2007*).

### Data processing

A total of 602 participants agreed to take part in our study, which took place between October 2020 and April 2021. The participants' explicit consent to data collection was obtained, and they also filled out a questionnaire on demographics and the BFI-10. Around 4.4 million records of app usage were acquired. Participants were excluded for having a low-quality sleep-wake cycle based on the usage of apps data if they: (i) had fewer

| Sequence ID | User Code | App Name | Start Time | End Time |
|---|---|---|---|---|
| 757 | etfgc{td\|83>6As``ge/nml | Maps | 23-Sep, 0:14:53 | 23-Sep, 0:14:55 |
| 758 | etfgc{td\|83>6As``ge/nml | Spotify | 23-Sep, 0:14:55 | 23-Sep, 0:15:33 |
| 759 | etfgc{td\|83>6As``ge/nml | Spotify | 23-Sep, 0:39:46 | 23-Sep, 0:39:49 |
| 760 | etfgc{td\|83>6As``ge/nml | Messenger | 23-Sep, 0:39:50 | 23-Sep, 0:40:10 |
| 761 | etfgc{td\|83>6As``ge/nml | Chrome | 23-Sep, 0:40:12 | 23-Sep, 0:44:02 |
| 762 | etfgc{td\|83>6As``ge/nml | Messenger | 23-Sep, 0:44:05 | 23-Sep, 0:44:45 |

**Figure 2** Example records of the usage of apps.

than seven full nights' worth of usage of apps; or (ii) had fewer than 16 h of recorded app usage within the 24 h maximum possible. The latter was made to exclude the case where data synchronization did not work correctly, *e.g.*, when the internet was off for a long time, or when the app was forced to restart. Figure 2 depicts an example of app usage. The usage of apps was processed to extract the human sleep-wake cycle patterns, as shown in Fig. 3. From such patterns, we were able to identify the nighttime period that had the most uninterrupted inactive time (free of smartphone use). We implemented interactive software to process the data and visualize the usage for each user and each day to enable manual annotation of sleeping and wake-up times. After the annotation, daily usage of apps, e-sleep duration, sleeping time, wake-up time, and sleep distraction were calculated from the objectively recorded usage data. We defined smartphone-distracted e-sleep as having been interrupted due to smartphone usage during sleeping hours. Finally, we created a dataset called the e-sleep dataset containing the e-sleep variables, demographics, and the BFI-10 score for each user ($N = 269$), as shown in the flowchart in Fig. 4. The student received the approval of the research ethics committee of the authors' institution.

## Analysis

From the smartphone usage data, we extract three continuous variables, *i.e.*, e-sleep duration (SDur), e-sleep distraction (SDist), and the daily average of smartphone usage (UsageOfAppsAvg), and four categorical variables, *i.e.*, e-sleep duration quality (SDurQ), e-sleep distraction category (SDurC), e-sleeping time (STime), e-wake-up time. Our continuous variable SDur represents the average daily duration of sleeping time. From SDur, a categorical variable called SDurQ was created. SDurQ represents the quality of sleep in terms of quantity, *i.e.*, poor, good, and over e-sleep. While sleeping, a smartphone could cause distraction. For example, users may check their phones in response to notifications, messages, or calls. They may also spend time on social media and messaging apps, where they are supposed to be continuing their sleep. We have visualized and scrutinized the smartphone usage data of each user to identify when such distraction happened and created the continuous variable SDist, which is the average daily time, in minutes, of sleep distraction caused by smartphones. From SDist, we created a categorical variable called SDistC consisting of two categories, *i.e.*, distracted and not-distracted. From the smartphone usage data, we extracted the sleeping time and the wake-up time of each user and processed the first and derived two categorical variables based on whether the user

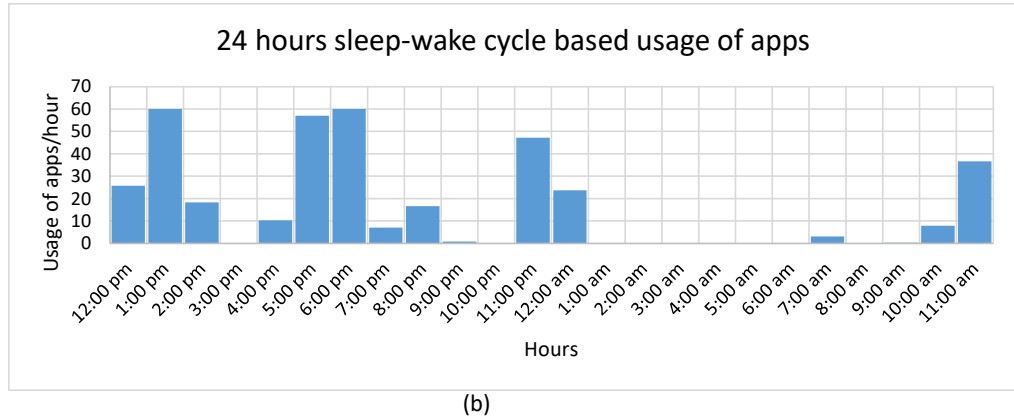

**Figure 3** (A) The human sleep-wake cycle over 24 h. (B) Smartphone usage over the human sleep-wake cycle along with lack of smartphone activities, *i.e.,* 12 a.m. to 7 a.m. This passive time is called "e-sleep" from the smartphone's perspective.

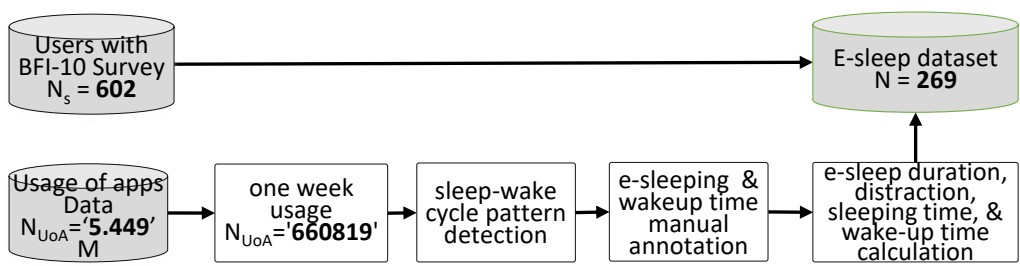

**Figure 4** **Flowchart of the data processing and formation of the e-sleep dataset.** The rectangle shape represents automatic processing, and the cylinder shape represents data storage.

sleeps at a regular time, *i.e.,* STime (early sleep, regular sleep, delayed sleep, and poor sleep), and processed the second in the same way regarding the time the user wakes up, *i.e.,* WTime (early wakeup, regular wakeup, delayed wakeup, and poor wakeup). Further explanation of these variables is shown in Table 1.

In this study, first, the Pearson's correlation analysis was conducted to explore the associations among variables. We then used multivariate linear regression models to predict SDur through PT, UsageOfAppsAvg of smartphone apps usage, age, and gender. While considering age and gender as factors, binomial logistic regression was used to estimate the impacts of PT and UsageOfAppsAvg on the likelihood that participants have had distracted sleep, measured through SDistC. Through PT, UsageOfAppsAvg, age, and

**Table 1  Description of the research variables.**

| S# | Variable | Description |
|---|---|---|
| 1. | E-Sleep Duration (SDur) | SDur is a continuous variable representing the daily average duration of the sleeping hours. We estimated the sleep duration of each night from the smartphone usage data and identified the non-usage period in minutes during the sleep-wake cycle. |
| 2. | E-Sleep Distraction (SDist) | SDist is a continuous variable representing the daily average duration in minutes of the distraction caused by smartphones during sleeping hours. |
| 3. | E-Sleep Duration Quality (**SDurQ**) | Based on the literature (*Hirshkowitz et al., 2015*), we create SDurQ categorical variable from SDur. SDurQ represents the sleep duration quality and consists of three categories, *i.e.*, poor e-sleep (if SDur is less than 7 h), good e-sleep (if SDur is between 7 h and 9 h), and over e-sleep (if SDur is greater than 9 h). |
| 4. | E-Sleep Distraction Category (SDistC) | Distraction happens when attention is given to the distracting object for more than eight seconds (Kelly Howard, 2019). Thus, based on SDist, we created SDistC categorical variable. SDistC has two categories, *i.e.*, not-distracted (if the distraction duration during sleeping hours is less than 0.13 min, *i.e.*, SDist <8 s), and distracted (if the distraction duration is equal or greater than 0.13 min, *i.e.*, SDist $\geq$ 8 s). |
| 5. | Daily Average usage of smartphone apps (UsageOfAppsAvg) | UsageOfAppsAvg is an independent continuous variable. We first calculated the smartphone usage of each participant for one week and then calculate the daily average usage in terms of minutes. UsageOfAppsAvg represents the daily average usage of smartphone apps. |
| 6. | E-Sleeping Time (STime) | Based on the literature (*Yan et al., 2021*), we create STime categorical variable with four categories, *i.e.*, early sleep (<10:00 PM), regular sleep (10:01 PM to 11:00 PM), delayed sleep (11:01 PM to 12:00 AM), and poor sleep (>12:01 AM). |
| 7. | E-Wake-up Time (WTime) | WTime is a categorical variable representing the average wakeup time of the participants. Based on the literature (*Yan et al., 2021*), we created a WTime variable with four categories, *i.e.*, early wakeup (<07:00 AM), regular wakeup (07:01 AM to 08:00 AM), delayed wakeup (08:01 AM to 09:00 AM), and poor wakeup (>09:01 AM). |

gender, a multinomial logistic regression analysis was used to predict E-Sleeping Time. Finally, PT, UsageOfAppsAvg, age, and gender were used to predict wakeup time in terms of early, regular, delayed, and poor wakeup time using multinomial logistic regression analysis.

**Table 2  Descriptive statistics for demography.**

| Variables | Frequency (269) | Percent |
|---|---|---|
| **Gender** | | |
| Male | 103 | 41.03 |
| Female | 148 | 55.01 |
| Others | 18 | 6.69 |
| **Age** | | |
| Emerging Adults (15–24) | 118 | 44.86 |
| Adults (25–64) | 145 | 55.13 |
| Missing | 6 | 2.23 |
| **Profession** | | |
| Students | 94 | 34.94 |
| Non-Students | 175 | 65.05 |

**Table 3  Descriptive statistics of e-sleep variables.** Values with ± indicated the standard deviation. All the variables are continuous and are calculated in terms of the daily average in minutes.

| | Male | Female | Students | Non-Students | Emerging Adults | Adults | Overall |
|---|---|---|---|---|---|---|---|
| SDur(M) | 465.61 (±75.87) | 469.47 (±73.81) | 471.18 (±76.22) | 468.19 (±73.07) | 467.63 (±73.83) | 469.27 (±74.58) | 469.24 (±74.06) |
| Poor sleep(M) | 374.44 (±36.50) | 370.36 (±45.01) | 379.89 (±35.22) | 367.25 (±44.57) | 379.45 (±39.73) | 364.30 (±42.90) | 371.53 (±33.67) |
| Good sleep(M) | 477.22 (±35.33) | 480.60 (±33.06) | 469.19 (±33.08) | 484.87 (±32.87) | 475.04 (±32.91) | 484.02 (±34.29) | 479.61 (±16.00) |
| Over sleep(M) | 576.19 (±44.44) | 572.70 (±572.70) | 582.47 (±46.23) | 564.11 (±25.97) | 574.52 (±42.03) | 567.98 (±30.33) | 571.67 (±36.44) |
| SDist(M) | 12.48 (±11.98) | 14.19 (±17.04) | 13.63 (±13.87) | 13.24 (±15.77) | 14.49 (±14.22) | 13.63 (±16.38) | 13.38 (±15.1) |
| UsageOfAppsAvg (M) | 286.58 (±154.6) | 317.52 (±149.97) | 339.26 (±21.96) | 286.01 (±20.03) | 337.28 (±163.77) | 280.32 (±138.3) | 304.62 (±151.3) |

## RESULTS

### Sample characteristics

Descriptive statistics of the participants' demographics are shown in Tables 2, 3 and 4. Of the 269 participants, 41.03% were male, and 44.86% were emerging adults between the ages of 15 and 24. The participants were from different professions, with 34.94% being students. The participants are from ten countries (India, the United States, the United Kingdom, Canada, Australia, Germany, the Netherlands, Brazil, France, and Sweden). Table 3 shows the mean average for men, women, students, non-students, adults, and emerging adults. The SDur average for males was 465.61 min and for females 469.47 min and found no significant differences between males and females.

As shown in Table 3, the average sleeping hours for males were 7.7, and for females were 7.8 h. Approximately 24%, 58%, and 17% of the users showed insufficient poor sleep, good sleep, and oversleeping in terms of duration, respectively, as shown in Fig. 4. 60.13% of

**Table 4  Descriptive statistics of sleeping and wakeup time categorical variables.**

| Variable | Male | Female | Students | Non-Students | Emerging Adults | Adults | Overall |
|---|---|---|---|---|---|---|---|
| Total | 41.03% | 58.96 | 34.94 | 65.05 | 43.86 | 53.90 | 100% |
| SDur | | | | | | | |
| Poor sleep | 27.18% | 23.64% | 23.40% | 29.05% | 26.27% | 23.44% | 24.16% |
| Good sleep | 56.31% | 60.13% | 56.38% | 70.94% | 55.93% | 60.68% | 58.73% |
| Over sleep | 16.50% | 16.21% | 20.21% | 18.24% | 17.79% | 15.86% | 17.10% |
| SDist | | | | | | | |
| Not Distracted | 16.50% | 18.91% | 15.95% | 18.85% | 16.10% | 18.37% | 20.07% |
| Distracted | 83.49% | 81.08% | 84.04% | 81.14% | 83.89% | 81.62% | 79.92% |
| STime | | | | | | | |
| Early Sleep | 5.8% | 5.40% | 6.38% | 4.57% | 5.08% | 4.82% | 5.20% |
| Regular Sleep | 21.35% | 16.21% | 12.76% | 23.42% | 11.86% | 24.82% | 19.70% |
| Delayed Sleep | 23.30% | 35.13% | 20.21% | 33.14% | 23.73% | 33.79% | 28.62% |
| Poor Sleep | 49.51% | 43.24% | 60.63% | 38.85% | 59.32% | 36.55% | 46.46% |
| WTime | | | | | | | |
| Early Wakeup | 25.24% | 16.21% | 13.82% | 22.28% | 13.55% | 23.44% | 17.10% |
| Regular Wakeup | 28.15% | 40.54% | 28.72% | 41.14% | 30.50% | 41.37% | 21.19% |
| Delayed Wakeup | 30.09% | 26.35% | 24.46% | 28.00% | 28.81% | 25.51% | 26.39% |
| Poor Wakeup | 16.50% | 16.89% | 32.97% | 8.57% | 27.11% | 9.65% | 35.31% |

**Notes.**
The percentage represents the percentage of the respective category. For example, out of 103 (41.03%) male participants, around 27% were poor sleepers.

females and 56.31% of males were found to be good sleepers, whereas 27.18% of females and 23.64% of males were found to be bad sleepers. 70.94% of non-students, 56.38% of students, 60.68% of adults, and 55.93% of emerging adults were observed as good sleepers. The average daily smartphone usage of the participant was around 304.62 min (5.07 h). Among them, the males were 286.58 min (4.77 h) and the females were 317.52 min (5.29 h). Whereas the average usage of the smartphone by students was 339.26 min (5.65 h) non-students had 286.01 min (4.76 h), emerging adults had 337.28 min (5.62), and adults had 280.32 min (4.67). Regarding sleep distraction, as shown in Table 3, the average sleeping distraction time was found to be 13.38 min, and approximately 80% of participants were classified as having distracted sleep because of smartphone interrupts.

As indicated in Table 4, approximately 5%, 20%, 29%, and 46% of the users showed early, regular, delayed, and poor sleeping time, respectively. Likewise, users with early wake-up were 5%, regular wake-up was 20%, delayed wake-up was 29%, and poor wake-up was 46%. Males (49.51%) were found to have poor sleeping time compared to females (43.24%). The percentage of students (60.63%) and emerging adults (59.32%) were high to have poor sleeping time compared to non-students (38.85%) and adults (36.55%), respectively. Regarding the wake-up time, more females were found to have regular wake-up times compared to males.

## Predicting sleep duration

To predict the impact of the PT and UsageOfAppsAvg on SDur, a multiple linear regression analysis was performed. First, the assumptions were checked and verified. The dependent

**Table 5  Correlation between BFI-10, daily average apps' usage, and e-sleep variables.**

| Variable | | 1 | 2 | 3 | 4 | 5 | 6 | 7 | 8 |
|---|---|---|---|---|---|---|---|---|---|
| | | | | **Pearson's Correlations** | | | | | |
| 1. Extraversion | Pearson's r | – | | | | | | | |
| | $p$-value | – | | | | | | | |
| 2. Agreeableness | Pearson's r | 0.111 | – | | | | | | |
| | $p$-value | 0.070 | – | | | | | | |
| 3. Conscientiousness | Pearson's r | 0.208*** | 0.063 | – | | | | | |
| | $p$-value | <.001 | 0.303 | – | | | | | |
| 4. Neuroticism | Pearson's r | −0.192** | −0.058 | −0.275*** | – | | | | |
| | $p$-value | 0.002 | 0.343 | <.001 | – | | | | |
| 5. Openness | Pearson's r | 0.139* | 0.128* | 0.096 | 0.062 | – | | | |
| | $p$-value | 0.023 | 0.037 | 0.115 | 0.314 | – | | | |
| 6. SDur | Pearson's r | 0.056 | 0.142* | 0.275*** | −0.119 | 0.092 | – | | |
| | $p$-value | 0.359 | 0.020 | <.001 | 0.051 | 0.132 | – | | |
| 7. SDist | Pearson's r | −0.084 | −0.045 | −0.196** | 0.088 | −0.063 | −0.233*** | – | |
| | $p$-value | 0.171 | 0.466 | 0.001 | 0.151 | 0.305 | <.001 | – | |
| 8. UsageOfAppsAvg | Pearson's r | −0.168** | −0.114 | −0.308*** | 0.204*** | −0.002 | −0.434*** | 0.321*** | – |
| | $p$-value | 0.006 | 0.062 | <.001 | <.001 | 0.979 | <.001 | <.001 | – |

**Notes.**
*$p < .05$
**$p < .01$
***$p < .001$

variable SDur was visually checked for being normally distributed. While performing the Shapiro–Wilk test, the $p$-value was not significant, *i.e.,* 0.247, which signifies the hypothesis of normality. There were no notable outliers in the SDur that significantly differed from the model based on the Standardized Residuals, which did not exceed −3.29 and 3.29. The collinearity statistics confirmed no multicollinearity among the variables. VIF values were less than 5 (for all predictors), and Tolerance values were more than 0.2. Pearson's correlation was also performed and showed no multicollinearity among the variables, as shown in Table 5. Durbin–Watson value, which should be between 1 and 3, was found to be 1.962, indicating the independence of predictors. The residuals' histograms were fairly normally distributed, the numbers did not exceed three, and the residuals' normality and homoskedasticity were fulfilled. The majority of the data points were on or near the line, according to the Q-Q Plot for residuals.

As shown in Table 6, PT, UsageOfAppsAvg, age, and gender were included in the multiple linear regression model as predictors and SDur as the dependent variable. The results revealed that the overall model contributed to the prediction of SDur with a variance of 26.2%. and adjusted $R^2$ of .238 (23.8%) where F (8, 240) = 10.658, and p <.001. We found that only UsageOfAppsAvg (B = −0.420, $p < 0.001$) and conscientiousness ($B = 0.186$, $p = 0.002$) were significant and positive predictors of SDur.

**Table 6  Main effects of multiple linear regression model in the prediction of SDur using personality traits, age, and gender as predictors.**

| Predictors | Standardized B | $t$ | $p$ | 95% CI | | Collinearity | |
|---|---|---|---|---|---|---|---|
| | | | | LB | UB | Tolerance | VIF |
| Constant | | 13.869 | <.001 | 393.407 | 523.661 | | |
| Openness | 0.068 | 1.169 | 0.244 | −1.740 | 6.821 | 0.919 | 1.088 |
| Extraversion | −0.087 | −1.493 | 0.137 | −6.895 | 0.949 | 0.901 | 1.110 |
| Agreeableness | 0.065 | 1.125 | 0.262 | −1.973 | 7.232 | 0.924 | 1.083 |
| Conscientiousness | 0.186 | 3.068 | 0.002 | 2.646 | 12.140 | 0.839 | 1.193 |
| Neuroticism | −0.022 | −0.369 | 0.713 | −4.606 | 3.153 | 0.841 | 1.189 |
| UsageOfAppsAvg | −0.420 | −7.079 | <.001 | −0.260 | −0.147 | 0.874 | 1.145 |
| Age | −0.058 | −1.009 | 0.314 | −25.576 | 8.246 | 0.925 | 1.081 |
| Gender | 0.059 | 0.997 | 0.320 | −8.643 | 26.344 | 0.883 | 1.132 |

**Notes.**
 $*p < .05$; $R^2 = 0.262$, Adjusted $R^2 = 0.238$, F(df) = 10.658 (8, 240), Durbin Watson = 1.954.

## Predicting sleep distraction

Binomial logistic regression analysis was performed to predict the effects of PT, and UsageOfAppsAvg on the likelihood that participants had distracted sleep, represented through variable SDistC (distracted, non-distracted), while considering age and gender as factors in the model. The logistic regression model was statistically significant, $\chi^2(390) = 57.278$, $p < .001$ between the outcome and the predictor variables. The model explained 17.8% (Nagelkerke $R^2$) of the variance in e-sleep distraction and correctly classified 63.0% of cases. Sensitivity and specificity were recorded at 64.5%, and 61.5%, respectively.

For distracted sleep relative to not distracted sleep, out of eight predictor variables, three were statistically significant: conscientiousness ($p < .001$, Exp(B) = 0.811), openness ($p = .011$, Exp(B) =0.867), and UsageOfAppsAvg ($p = .002$, Exp(B) = 1.002), as shown in Table 7. While looking at the value of the Exp(B), if a subject were to increase the conscientiousness score by one point, the log-odds of preferring not-distracted to distracted sleep would be expected to increase by 0.811 units while holding all other variables in the model constant. This suggests that lower conscientiousness levels are significantly related to an increased probability of having distracted sleep. If a subject were to increase the openness score by one point, the log-odds of preferring not-distracted to distracted sleep would be expected to increase by 0.867 units while holding all other variables in the model constant. This suggests that higher openness levels are significantly related to decreased probability of having distracted sleep. Increasing UsageOfAppsAvg is related to a significantly increased probability of distracted sleep. The agreeableness, extraversion, neuroticism, and factors (age and gender) were not found statistically significant. This relation of conscientiousness, openness, and UsageOfAppsAvg with the SDistC is visualized in Fig. 5.

## Predicting sleeping time

Multinomial logistic regression analysis was performed to predict e-sleeping time in terms of early, regular, delayed, and poor sleeping time through PT, UsageOfAppsAvg, age, and gender. The full model, model fitting, was statistically significant (*i.e.*, $p < .001$) and

**Table 7** Logistic regression model predicting the likelihood of SDistC based on personality traits, UsageOfAppsAvg, Age, and Gender.

| | Estimate (B) | SE | Exp (B) | Z | Wald Test | | | 95% CI (odds ratio scale) | |
| --- | --- | --- | --- | --- | --- | --- | --- | --- | --- |
| | | | | | Wald Statistic | df | p | LB | UB |
| (Intercept) | 0.250 | 0.802 | 1.283 | 0.311 | 0.097 | 1 | 0.756 | −1.322 | 1.821 |
| Extraversion | 0.070 | 0.050 | 1.073 | 1.399 | 1.959 | 1 | 0.162 | −0.028 | 0.169 |
| Agreeableness | 0.106 | 0.058 | 1.112 | 1.822 | 3.321 | 1 | 0.068 | −0.008 | 0.221 |
| Conscientiousness | −0.210 | 0.062 | 0.811 | −3.360 | 11.293 | 1 | <.001 | −0.332 | −0.087 |
| Neuroticism | 0.088 | 0.051 | 1.092 | 1.717 | 2.947 | 1 | 0.086 | −0.013 | 0.189 |
| Openness | −0.143 | 0.056 | 0.867 | −2.535 | 6.426 | 1 | 0.011 | −0.253 | −0.032 |
| UsageOfAppsAvg | 0.002 | 0.001 | 1.002 | 3.054 | 9.326 | 1 | 0.002 | 0.001 | 0.004 |
| Age (Adults) | 0.019 | 0.358 | 1.019 | 0.053 | 0.003 | 1 | 0.957 | −0.682 | 0.720 |
| Gender (Female) | −0.395 | 0.339 | 0.674 | −1.165 | 1.357 | 1 | 0.244 | −1.059 | 0.270 |
| Age (Adults) * Gender (Female) | 0.096 | 0.442 | 1.101 | 0.218 | 0.047 | 1 | 0.828 | −0.770 | 0.962 |

**Notes.**
SDistC level 'Distracted' coded as class 1.

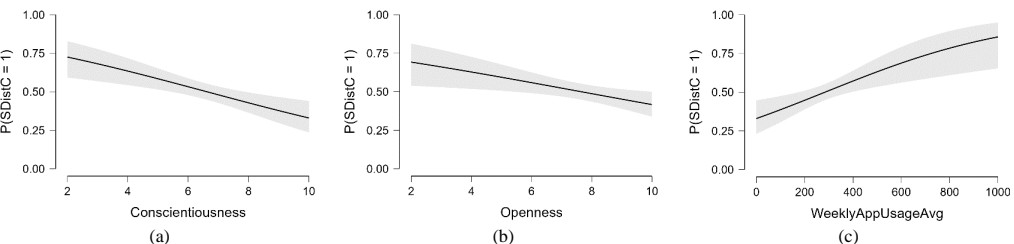

**Figure 5** **(A) An increase in conscientiousness decreases the probability of having an e-sleep distraction, (B) an increase in openness decreases the probability of having an e-sleep distraction, (C) and an increase in usage of apps increase the sleep distraction.**

predicted the dependent variable better than the intercept-only model alone. The pseudo-R-square measures, *i.e.,* Cox and Snell, Nagelkerke, and McFadden were found to be 0.360 (36.0%), 0.384 (38.4%), and 0.161 (16.1%), respectively. The Likelihood ratio test (Table 8) shows the contribution of each variable to the model. Conscientiousness ($p < .001$), UsageOfAppsAvg ($p < 0.001$), age ($p < 0.001$), and gender ($p = 0.006$) contributed significantly, but extraversion, agreeableness, and neuroticism did not.

For early sleep relative to regular sleep, as shown in Table 9, conscientiousness (Exp(B) = 1.472 (95% CI 1.232 to 1.759), $p < 0.001$, Wald = 18.115), and age (Exp(B) = 2.779 (95% CI 1.508 to 5.121), $p < 0.001$, Wald = 10.740) were statistically significant. The multinomial log-odds of choosing regular to early sleep should fall by 0.472 units whenever a subject's conscientiousness score increases by one point, according to the conscientiousness Exp(B) value, while maintaining all other model variables constant. More generally, it can be inferred that if a subject were to increase their conscientiousness score, we would expect them to be more likely in early sleep over regular sleep. For age (Exp(B) = 2.779, $p < 0.001$), emerging adults compared to adults were more likely to be early sleepers. Conversely, we

**Table 8  Likelihood Ratio Tests: the contribution of personality traits, UsageOfAppsAvg, age, and gender to the model to predict STime.**

| Effect | Model fitting criteria | Likelihood Ratio Tests | | |
|---|---|---|---|---|
| | −2 Log Likelihood of Reduced Model | Chi-Square | df | Sig. |
| Intercept | 1074.060 | .000 | 0 | . |
| Extraversion | 1075.657 | 1.596 | 3 | .660 |
| Agreeableness | 1078.388 | 4.328 | 3 | .228 |
| Conscientiousness | 1102.990 | 28.930 | 3 | <.001 |
| Neuroticism | 1077.472 | 3.411 | 3 | .332 |
| Openness | 1074.649 | .589 | 3 | .899 |
| UsageOfAppsAvg | 1178.909 | 104.848 | 3 | <.001 |
| Age | 1090.142 | 16.082 | 3 | .001 |
| Gender | 1086.606 | 12.546 | 3 | .006 |

can say that adults were more prone to be regular sleepers, Exp(B) = 0.359 (given by the reciprocal of 2.779).

For delayed relative to regular sleeping time, the UsageOfAppsAvg (Exp(B) = 1.008 (95% CI 1.005 to 1.011), $p < 0.001$, Wald =29.187), and gender (Exp(B) = 0.439 (95% CI .241 to 0.797), $p = 0.007$, Wald = 7.309) were found statistically significant. The multinomial log-odds of preferring delayed to regular sleeping time would be expected to decrease by 0.008 units while holding all other variables in the model constant, according to the UsageOfAppsAvg's Exp(B) value. This is because if a subject were to raise their UsageOfAppsAvg score by one point. More generally, it can be inferred that if a subject were to increase their UsageOfAppsAvg score, we would expect them to be more likely in the delayed sleeping time compared to regular sleep. For gender (Exp(B) = .439, $p = 0.007$), males compared to females were more likely to be delayed sleepers. Conversely, we can say that females were more prone to be regular sleepers, Exp(B) = 0.359 (given by the reciprocal of 2.779).

Likewise, for poor relative to regular sleeping time, the UsageOfAppsAvg (Exp(B) = 1.011 (95% CI 1.008 to 1.014), $p < 0.001$, Wald = 52.229), and age (Exp(B) = 2.449 (95% CI 1.292 to 4.645), $p = 0.006$, Wald = 7.527) were found statistically significant. The Exp(B) value for UsageOfAppsAvg indicates that if a subject were to increase the UsageOfAppsAvg score by one point, the multinomial log-odds of preferring poor to regular sleeping time would be expected to increase by 0.011 units while holding all other variables in the model constant. For age, emerging adults compared to adults were more likely to fall in the poor sleeping time category compared to regular sleeping.

## Predicting wakeup time

Multinomial logistic regression analysis was performed to predict wakeup time in terms of early, regular, delayed, and poor wakeup time through PT, UsageOfAppsAvg, age, and gender. The full model was statistically significant (*i.e.*, $p < .001$) and predicted the dependent variable better than the intercept-only model alone. The pseudo-R-square measures, *i.e.*, Cox and Snell, Nagelkerke, and McFadden were found 0.254 (25.4%), 0.273 (27.3%), and 0.110 (11.0%), respectively. The Likelihood Ratio Test (Table 10) shows

**Table 9** Parameter estimates of the multinomial logistic regression model while showing the relationships of sleep time categories (STime), *i.e.,* early sleep, regular sleep, delayed sleep, and poor sleep using personality traits, UsageOfAppsAvg, age, and gender as predictors.

| STime[a] | | B | Std. Error | Wald | df | Sig. | Exp(B) | 95% CI for Exp(B) | |
|---|---|---|---|---|---|---|---|---|---|
| | | | | | | | | LB | UB |
| Early Sleep | Intercept | −1.511 | 1.229 | 1.511 | 1 | .219 | | | |
| | Extraversion | −.083 | .068 | 1.476 | 1 | .224 | .920 | .805 | 1.052 |
| | Agreeableness | −.101 | .077 | 1.738 | 1 | .187 | .904 | .777 | 1.051 |
| | Conscientiousness | .387 | .091 | 18.115 | 1 | .000 | 1.472 | 1.232 | 1.759 |
| | Neuroticism | −.065 | .068 | .932 | 1 | .334 | .937 | .821 | 1.070 |
| | Openness | .014 | .074 | .036 | 1 | .850 | 1.014 | .877 | 1.172 |
| | UsageOfAppsAvg | .000 | .002 | .000 | 1 | .993 | 1.000 | .997 | 1.003 |
| | [Age= Emerging Adults] | 1.022 | .312 | 10.740 | 1 | .001 | 2.779 | 1.508 | 5.121 |
| | [Gender=Male] | −.204 | .302 | .455 | 1 | .500 | .816 | .451 | 1.475 |
| Delayed Sleep | Intercept | −1.643 | 1.211 | 1.842 | 1 | .175 | | | |
| | Extraversion | −.020 | .065 | .094 | 1 | .759 | .980 | .864 | 1.113 |
| | Agreeableness | .054 | .080 | .461 | 1 | .497 | 1.056 | .903 | 1.234 |
| | Conscientiousness | .111 | .085 | 1.703 | 1 | .192 | 1.117 | .946 | 1.319 |
| | Neuroticism | −.121 | .066 | 3.354 | 1 | .067 | .886 | .778 | 1.009 |
| | Openness | −.028 | .073 | .147 | 1 | .702 | .972 | .843 | 1.122 |
| | UsageOfAppsAvg | .008 | .001 | 29.187 | 1 | .000 | 1.008 | 1.005 | 1.011 |
| | [Age= Emerging Adults] | .275 | .309 | .793 | 1 | .373 | 1.317 | .719 | 2.412 |
| | [Gender=Male] | −.824 | .305 | 7.309 | 1 | .007 | .439 | .241 | .797 |
| Poor Sleep | Intercept | −2.097 | 1.326 | 2.500 | 1 | .114 | | | |
| | Extraversion | −.035 | .070 | .245 | 1 | .620 | .966 | .842 | 1.108 |
| | Agreeableness | .008 | .086 | .008 | 1 | .930 | 1.008 | .851 | 1.194 |
| | Conscientiousness | −.059 | .093 | .403 | 1 | .526 | .943 | .786 | 1.131 |
| | Neuroticism | −.079 | .074 | 1.133 | 1 | .287 | .924 | .798 | 1.069 |
| | Openness | −.041 | .079 | .272 | 1 | .602 | .960 | .822 | 1.120 |
| | UsageOfAppsAvg | .011 | .002 | 52.229 | 1 | .000 | 1.011 | 1.008 | 1.014 |
| | [Age= Emerging Adults] | .896 | .327 | 7.527 | 1 | .006 | 2.449 | 1.292 | 4.645 |
| | [Gender=Male] | .118 | .317 | .139 | 1 | .709 | 1.126 | .605 | 2.096 |

**Notes.**
[a]The reference category is: Regular Sleep.

the contribution of each variable to the model. Extraversion ($p < .001$), Agreeableness ($p < .001$), Conscientiousness ($p = .022$), Neuroticism ($p = .003$), age ($p < .001$), and gender ($p < .001$) contributed significantly, but not openness, and UsageOfAppsAvg.

For "early wakeup" relative to "regular wakeup", as shown in Table 11, the extraversion (Exp(B) = 1.228 (95% CI 1.084 to 1.391), $p < 0.001$, Wald = 10.44), agreeableness (Exp(B) = 0.678 (95% CI .586 to 0.784), $p < 0.001$, Wald = 27.20), conscientiousness (Exp(B) = 1.183 (95% CI 0.740 to 1.004), $p = 0.057$, Wald = 3.62), neuroticism (Exp(B) = 1.183 (95% CI 1.041 to 1.345), $p = 0.010$, Wald = 6.59), and gender (Exp(B) = 3.913 (95% CI 2.233 to 6.857), $p < 0.001$, Wald = 22.71) were statistically significant. The Exp(B) value for extraversion indicates that if a subject were to increase the extraversion score by one point, the multinomial log-odds of preferring "regular wakeup" to "early wakeup"

**Table 10  Likelihood Ratio Tests: the contribution of personality traits, UsageOfAppsAvg, age, and gender to the model to predict WTime.**

| Effect | Model Fitting Criteria | Likelihood Ratio Tests | | |
|---|---|---|---|---|
| | −2 Log Likelihood of Reduced Model | Chi-Square | df | Sig. |
| Intercept | 1092.085 | .000 | 0 | . |
| Extraversion | 1109.182 | 17.097 | 3 | .001 |
| Agreeableness | 1122.982 | 30.897 | 3 | .000 |
| Conscientiousness | 1101.751 | 9.666 | 3 | .022 |
| Neuroticism | 1106.130 | 14.045 | 3 | .003 |
| Openness | 1097.752 | 5.667 | 3 | .129 |
| UsageOfAppsAvg | 1096.521 | 4.436 | 3 | .218 |
| Age | 1118.794 | 26.709 | 3 | .000 |
| Gender | 1116.417 | 24.332 | 3 | .000 |

would be expected to decrease by 0.228 units while holding all other variables in the model constant. The Exp(B) value for neuroticism indicates that if a subject were to increase the neuroticism score by one point, the multinomial log-odds of preferring "regular wakeup" to "early wakeup" would be expected to decrease by 0.183 units while holding all other variables in the model constant. More generally, it can be inferred that if a subject were to increase their extraversion and/or neuroticism score, we would expect them to be more likely in the early wakeup over regular wakeup. The Exp(B) value for conscientiousness indicates that if a subject were to increase the conscientiousness score by one point, the multinomial log-odds of preferring "regular wakeup" to "early wakeup" would be expected to increase by 0.862 units. The Exp(B) value for agreeableness indicates that if a subject were to increase the agreeableness score by one point, the multinomial log-odds of preferring "regular wakeup" to "early wakeup" would be expected to increase by 0.678 units. For males relative to females, the relative risk of preferring "early wakeup" to "regular wakeup" would be expected to increase by a factor of 0.913 given the other variables in the model are held constant. In other words, males are more likely than females to prefer "early wakeup" to "regular wakeup", Exp(B) = 0.255 (given by the reciprocal of 3.913). The multinomial log-odds of choosing "regular wakeup" over "early wakeup" is projected to fall by 0.183 units if a subject's neuroticism score were to rise by one point, according to the neuroticism score's Exp(B) value, while holding all other model variables constant. More broadly, it may be deduced that we would anticipate a subject to be more likely in the early wakeup than ordinary wakeup if their extraversion and/or neuroticism scores increased. According to the conscientiousness score's Exp(B) value, if a subject's conscientiousness level were to rise by one point, the multinomial log-odds that they would choose "normal wakeup" over "early wakeup" were predicted to rise by 0.862 units. According to the agreeableness score's Exp(B) value, the multinomial log-odds of choosing "normal wakeup" over "early wakeup" should rise by 0.678 units if the agreeableness score were raised by one point. Given that the other factors in the model are maintained constant, it would be predicted that males are more likely than females to choose "early wakeup" over "normal wakeup". In other words, Exp(B) = 0.255 indicates that males are more likely than females to choose "early wakeup" over "normal wakeup" (given by the reciprocal of 3.913).

**Table 11** Parameter estimates of the multinomial logistic regression model while showing the relationships of wakeup time categories (WTime), *i.e.*, early wakeup, regular wakeup, delayed wakeup, and poor wakeup using personality traits, UsageOfAppsAvg, age, and gender as predictors.

| WTime[a] | | B | Std. Error | Wald | df | Sig. | Exp(B) | 95% CI for Exp(B) | |
|---|---|---|---|---|---|---|---|---|---|
| | | | | | | | | LB | UB |
| Early Wakeup | Intercept | −.622 | 1.155 | .290 | 1 | .590 | | | |
| | Extraversion | .205 | .064 | 10.446 | 1 | .001 | 1.228 | 1.084 | 1.391 |
| | Agreeableness | −.389 | .075 | 27.207 | 1 | .000 | .678 | .586 | .784 |
| | Conscientiousness | −.149 | .078 | 3.624 | 1 | .057 | .862 | .740 | 1.004 |
| | Neuroticism | .168 | .065 | 6.599 | 1 | .010 | 1.183 | 1.041 | 1.345 |
| | Openness | .054 | .069 | .618 | 1 | .432 | 1.056 | .922 | 1.208 |
| | WeeklyAppUsageAvg | .002 | .001 | 2.300 | 1 | .129 | 1.002 | 1.000 | 1.004 |
| | [Age= Emerging Adults] | .034 | .282 | .014 | 1 | .905 | 1.034 | .595 | 1.799 |
| | [Gender=Male] | 1.364 | .286 | 22.712 | 1 | .000 | 3.913 | 2.233 | 6.857 |
| Delayed Wakeup | Intercept | 1.732 | 1.111 | 2.430 | 1 | .119 | | | |
| | Extraversion | .118 | .062 | 3.548 | 1 | .060 | 1.125 | .995 | 1.271 |
| | Agreeableness | −.232 | .075 | 9.583 | 1 | .002 | .793 | .685 | .919 |
| | Conscientiousness | −.206 | .077 | 7.240 | 1 | .007 | .814 | .700 | .946 |
| | Neuroticism | .119 | .061 | 3.755 | 1 | .053 | 1.127 | .999 | 1.271 |
| | Openness | −.112 | .067 | 2.832 | 1 | .092 | .894 | .785 | 1.019 |
| | WeeklyAppUsageAvg | −.001 | .001 | .404 | 1 | .525 | .999 | .997 | 1.001 |
| | [Age= Emerging Adults] | .246 | .274 | .806 | 1 | .369 | 1.279 | .747 | 2.191 |
| | [Gender=Male] | .462 | .281 | 2.703 | 1 | .100 | 1.587 | .915 | 2.753 |
| Poor Wakeup | Intercept | 1.674 | 1.376 | 1.479 | 1 | .224 | | | |
| | Extraversion | −.082 | .077 | 1.137 | 1 | .286 | .921 | .792 | 1.071 |
| | Agreeableness | −.205 | .092 | 4.911 | 1 | .027 | .815 | .680 | .977 |
| | Conscientiousness | −.204 | .095 | 4.603 | 1 | .032 | .815 | .676 | .982 |
| | Neuroticism | −.095 | .075 | 1.592 | 1 | .207 | .909 | .784 | 1.054 |
| | Openness | .001 | .084 | .000 | 1 | .991 | 1.001 | .849 | 1.180 |
| | WeeklyAppUsageAvg | .000 | .001 | .142 | 1 | .706 | 1.000 | .998 | 1.003 |
| | [Age= Emerging Adults] | 1.594 | .342 | 21.760 | 1 | .000 | 4.922 | 2.520 | 9.615 |
| | [Gender=Male] | .537 | .324 | 2.746 | 1 | .097 | 1.710 | .907 | 3.226 |

**Notes.**
[a]The reference category is: Regular wakeup.

For "delayed wakeup" relative to "regular wakeup", the second section of Table 11, agreeableness (Exp(B) = 0.793 (95% CI 0.685 to 0.919), $p = 0.002$, Wald = 9.583), conscientiousness (Exp(B) = 0.814 (95% CI 0.700 to 0.946), $p = 0.007$, Wald = 7.240), and neuroticism (Exp(B) = 1.127 (95% CI 0.999 to 1.271), $p = 0.053$, Wald = 3.755) were statistically significant. The Exp(B) value for agreeableness indicates that if a subject were to increase the agreeableness score by one point, the multinomial log-odds of preferring "regular wakeup" to "delayed wakeup" would be expected to decrease by 0.793 units while holding all other variables in the model constant. The Exp(B) value for conscientiousness indicates that if a subject were to increase the conscientiousness score by one point, the multinomial log-odds of preferring "regular wakeup" to "delayed wakeup" would be expected to decrease by 0.183 units while holding all other variables in the model

constant. It can be inferred that if a subject were to increase their agreeableness and/or conscientiousness score, we would expect them to be more likely in the "delayed wakeup" over "regular wakeup". The Exp(B) value for neuroticism indicates that if a subject were to increase the neuroticism score by one point, the multinomial log-odds of preferring "regular wakeup" to "delayed wakeup" would be expected to increase by 0.862 units.

The third section of Table 11 shows the relation of "poor wakeup" to "regular wakeup". Agreeableness (Exp(B) = 0.815 (95% CI .680 to.977), $p = 0.027$, Wald = 4.911), conscientiousness (Exp(B) = 0.815 (95% CI 0.676 to 0.982), $p = .032$, Wald = 4.603), and age (Exp(B) = 4.922 (95% CI 2.520 to 9.615), $p < 0.001$, Wald = 21.760) were statistically significant. The Exp(B) value for agreeableness indicates that if a subject were to increase the agreeableness score by one point, the multinomial log-odds of preferring "regular wakeup" to "poor wakeup" would be expected to decrease by 0.815 units while holding all other variables in the model constant. The Exp(B) value for conscientiousness indicates that if a subject were to increase the conscientiousness score by one point, the multinomial log-odds of preferring "regular wakeup" to "delayed wakeup" would be expected to decrease by 0.815 units while holding all other variables in the model constant. It can be inferred that if a subject were to increase their agreeableness and/or conscientiousness score, we would expect them to be more likely in the "regular wakeup" over "poor wakeup". For age, emerging adults compared to adults were more likely to fall in the "poor wakeup" time category compared to "regular wakeup".

## DISCUSSION

Personality traits (PT) are thought to influence a wide range of human behaviors (*Amichai-Hamburger, 2002*; *Meston, Trapnell & Gorzalka, 1996*). The association between PT and sleep quality has been researched over the years. The authors in *Kaur & Kaur (2021)* researched the association between PT, excessive internet usage, and sleep-wake patterns. The study did not include other sleep variables, *i.e.,* sleep duration, and distraction. Likewise, the association between PT and sleep quality was explored by *Gamaldo et al. (2020)*, *Hintsanen et al. (2014)*, *Kheirandish et al. (2020)*, *Lane et al. (2021)* and *Soehner, Kennedy & Monk (2007)*. The findings of the existing literature had limitations. Studies are either cross-sectional or subjective relying on methods like surveys for collecting variables around sleep and technology usage. For example, most of the existing research utilized self-report measures for sleep quality, such as the Pittsburgh Sleep Quality Index (PSQI) calculates a composite score for each participant based on their ratings of their subjective sleep quality, sleep latency, sleep length, habitual daytime dysfunction, sleep disruptions, and usage of sleeping pills (*Cellini, Duggan & Sarlo, 2017*; *Duggan et al., 2014*; *Gray & Watson, 2002*; *Huang et al., 2016*; *Kim et al., 2015*; *Williams & Moroz, 2009*). A comparison with related work is shown in Table 12.

We overcome some of the limitations of existing research work by relying on objective monitoring of smartphone usage for one week. We then study the role of PT and objectively recorded smartphone usage in the prediction of sleep variables, which is a unique combination of variables that have been studied earlier. In this study, sleep variables

**Table 12 Comparison with related work.**

| Ref. | PT Questionnaire | Sleep Quality | | Smartphone Usage | | Participants & Sample Size | SWC | Description |
|---|---|---|---|---|---|---|---|---|
| | | **Self-reports Questionnaire** | **Objective Measure** | **Questionnaire** | **Objective Assessments** | | | |
| *Kaur & Kaur (2021)* | - EPQ-R (*Lane et al., 2021*) | - Sleep-Wake Pattern Assessment (*Melnikov et al., 1999*) | no | Internet Addiction Test (IAT, *Young, 1998*) | no | - Adolescents Aged = 16–18<br>- N = 350 | no | Studies PT & sleeping and wakeup patterns |
| *Hintsanen et al. (2014)* | - NEO-FFI questionnaire (*McCrae & Costa, 1989*) | - Sleep Duration<br>- Sleep Deficiency | no | no | no | - Aged = 31–45<br>- N = 1623 | no | Associations between PT and sleep duration, sleep deficiency & sleep problems |
| *Stephan et al. (2018)* | - Midlife Development Inventory (*Lachman & Weaver, 1997*) | - Sleeping Trouble<br>- Wakeup Trouble | no | no | no | - N = 22,000<br>- Aged 30 to 107 | no | Association between PT & SQ in 4 samples of middle-aged and older adults |
| *Kim et al. (2015)* | - NEO-PI-R and NEO (*McCrae & Costa, 1989*) | - PSQI (*Smyth, 1999*) | no | no | no | - Korean Men & Women<br>- N = 1406 | no | Association between PT & SQ in Young Korean Women |
| *Soehner, Kennedy & Monk (2007)* | - Eysenck Personality Inventory (*Eysenck & Eysenck, 1965*)<br>- The Attitude to Life Questionnaire (*Monk et al., 2001*) | - The Sleep Timing Questionnaire (*Monk et al., 2003*),<br>- The Composite Scale of Morningness (*Smith, Reilly & Midkiff, 1989*),<br>- PSQI (*Buysse et al., 1989*) | no | no | | - University Alumni,<br>- N = 225<br>- Aged = 23–48 | no | PT Correlation with Sleep-Wake Variables |
| *Gamaldo et al. (2020)* | - Big Five Inventory (*John, Naumann & Soto, 2008*) | - PSQI (*Buysse et al., 1989*) | no | no | no | - N = 112<br>- Aged = 55–86 | no | To explore the relationship between PT and sleep in community-dwelling older Blacks |

*(continued on next page)*

| Ref. | PT Question-naire | Sleep Quality | | Smartphone Usage | | Participants & Sample Size | SWC | Description |
|------|-------------------|---------------|--|------------------|--|---------------------------|-----|-------------|
| | | Self-reports Question-naire | Objective Measure | Question-naire | Objective As-sessments | | | |
| *Lane et al. (2021)* | - Tri-Dimensional Personality Questionnaire (*Cloninger, Przybeck & Svrakic, 1991*) | - Beck De-pression (*Beck & Steer, 1984*) and depres-sion Inventory (*Che et al., 2006*) | no | Smartphone Addiction In-ventory (*Lin et al., 2014*) | no | - Univer-sity Students - N = 422 | no | Association between PT & smartphone addiction & its effects on sleep distur-bance. |
| Our Work | BFI-10 (*Rammstedt & John, 2007*) | no | yes | no | yes | - From 10 countries - Aged = 15–65 N = 269 | yes | To assess the role of PT and smartphone usage in sleep quality |

**Notes.**

SWC, Sleep-wake Cycle.

were also extracted from the smartphone usage data based on the human sleep-wake cycle, *i.e.,* more objectively than relying on personal recall of them.

## Sleep duration prediction

As indicated in Table 3, the average sleeping hours for males were 7.7, and for females were 7.8 h. Approximately 24%, 58%, and 17% of our participants showed poor sleep, good sleep, and oversleeping in terms of duration, respectively (Table 4). Previous research found a 42.4% to 60% prevalence of poor sleep duration (*Ayala et al., 2017*; *Costa, Patrão & Machado, 2019*; *Pensuksan et al., 2016*; *Yazdi et al., 2016*). This difference might be due to recall bias in self-reported sleep data in previous research, or to the fact that we collected our data during the COVID-19 period when they had fewer sleep restrictions, *e.g.,* on waking up time (*Yuksel et al., 2021*). Another reason might be our reliance on smartphone usage to estimate sleep time, whereas people may be away from their phones but not asleep. Females (60.13%) were found to be good sleepers compared to males (56.31%), which is consistent with the findings of *Yazdi et al. (2016)*. Non-students (70.94%) and adults (60.68%) were found to be good sleepers compared to students (56.38%) and emerging adults (55.93%). It shows that non-students and adults can keep themselves away from their smartphones during the recommended sleep time.

In the correlation analysis, Table 5, the personality traits of agreeableness ($r = 0.14$; $p < 0.05$) and conscientiousness ($r = 0.27$; $p < 0.001$) were found to have a significant positive relationship with sleep duration. In addition to the correlational analysis, the regression analysis revealed the significant contribution of conscientiousness and usage of smartphones in the prediction of sleep duration, as shown in Table 6. These findings are consistent with those in *Stephan et al. (2018)*, *i.e.,* low conscientiousness was associated with a worsening of quality sleep duration. Our second finding, *i.e.,* smartphone usage related to sleep duration, supports the findings in *Cain & Gradisar (2010)*, *Mohammadbeigi et al.*

*(2016)*, *Tamura et al. (2017)* and *Orzech et al. (2016)*, *i.e.,* participants who spent more time using digital media, had less sleep duration.

## Sleeping distraction prediction

Regarding sleep distraction (Table 3), the average sleeping distraction time was found to be 13.38 min, and approximately 80% of participants had distracted sleep due to smartphone interrupts. The authors in *Jniene et al. (2019)* conducted a study before COVID-19 and found that 60% of the participants had sleep distractions due to smartphone usage. The distraction was almost uniform across gender, profession, and age. Table 6 shows that the personality trait of conscientiousness ($r = -0.196$; $p < 0.001$) had a significant negative relationship with sleep distraction due to smartphone usage. Regression analysis revealed that conscientiousness, openness, and smartphone usage amount were all significant predictors of sleep distraction, as shown in Table 7. This association between smartphone overuse and sleep distraction is consistent with findings in *Cain & Gradisar (2010)* and *Foundation (2011)*.

Conscientious people are accountable, trustworthy, motivated, and may be driven to get enough sleep to get more done throughout the day. Higher levels of conscientiousness (self-control) may enable people to better manage their impulses, refrain from excessive smartphone use and other sleep-impairing activities, and obtain a decent night's sleep. Most studies have also found a relationship between conscientiousness and better sleep quality (*Duggan et al., 2014*; *Gray & Watson, 2002*; *Hintsanen et al., 2014*; *Kim et al., 2015*; *Williams & Moroz, 2009*), although two studies did not find support for this association (*Allen, Magee & Vella, 2016*; *Cellini, Duggan & Sarlo, 2017*). Openness has generally been found to be unrelated to global measures of sleep quality (*Duggan et al., 2014*; *Hintsanen et al., 2014*; *Huang et al., 2016*; *Kim et al., 2015*), but one study found that higher openness is related to poorer sleep quality (*Allen, Magee & Vella, 2016*). Unlike existing work, we found that lower openness is related to higher sleep distraction when considering smartphone usage. In this context, the study by *Takao (2014)* found that openness has a negative correlation with problematic smartphone use. Moreover, smartphone overuse is associated with poor sleep quality (*Yang et al., 2020*). People with high openness are curious about new things, ideas and people, eager to learn, and have new experiences. Consequently, a high level of openness may be less dependent on smartphones, which could lead to quality sleep. We also found evidence that agreeableness ($p < .068$, Exp(B) $= 1.112$) is almost significant and associated with low peer influence resistance, and this can interpret why people distract from their sleep, perhaps to check and reply to messages. However, this requires further analysis of the apps that were used when a person was distracted.

## Sleeping and wakeup time prediction

Regression analysis revealed that conscientiousness, smartphone usage, age, and gender have a significant contribution to the prediction of sleeping time. A high level of conscientiousness could lead to early or /regular sleeping time. We discovered that increased smartphone usage is associated with less sleep, which is not surprising given past research linking digital media usage and sleep (*Orzech et al., 2016*; *Smyth, 2007*; *Wolfe*

*et al., 2014*). Emerging adults with a high level of conscientiousness would prefer early sleep compared to adults. Emerging adults with a high level of smartphone usage would fall into poor sleep compared to adults.

Regarding waking-up time prediction, extraversion, neuroticism, conscientiousness, agreeableness, age, and gender were found to be significant predictors. Smartphone daily average usage has shown no relation with wake-up time. However, an increase in smartphone usage leads to poor sleep duration, distraction, and sleeping time. In the regression analyses, conscientiousness remained a significant predictor of sleep quality in terms of duration. High levels of conscientiousness and agreeableness resulted in people being more likely to have regular wake-up times, whereas increasing extraversion and neuroticism led to poor wake-up time. In the regression analysis, when analyzing the impact of PT and smartphone usage on WTime, we found that males prefer early wake-up compared to females, and emerging adults were more likely to fall into poor wake-up time compared to adults.

These results demonstrate that the extraversion personality trait has an impact on controlling and maintaining a wake-up schedule. This is aligned with recent research that looked at the vital role of extraversion in morningness (*Pljusnin, 1993*; *Tankova, Adan & Buela-Casal, 1994*). The results are also consistent with Eyesenck's personality theory, which states that extraverts focus their attention on the outer world. They most likely pay closer attention, actively seek out external stimuli, and also have a more positive perspective on the situation, and perceive less threat in what is happening (*Rosswurm, Pierson & Woodward, 2001*). Positive affect is more prevalent in people with higher extraversion traits. According to empirical data, positive people have more stable circadian rhythms (*Kim, 2013*). Extroverts' reactions in a given scenario are significantly influenced by this positive psychological condition. Therefore, even when using a smartphone, users with higher extraversion were able to maintain healthy sleep habits.

We also found that an increase in neuroticism leads to delayed wake-up. People with high neuroticism do not have a specific wake and sleep times and are unable to sleep for an adequate amount of time (*Kaur & Kaur, 2021*). Similar results were achieved by *Kaur & Kaur (2021)*, who reported that internet users who possess the neuroticism personality trait are more likely to have wake-up delays. According to *Eysenck (1991)* and *Muris et al. (2005)*, neurotics have higher levels of anxiety, which may be related to higher levels of arousal that cause them to stay awake longer before falling asleep. Additionally, those who exhibit this tendency are susceptible to daily stressors (*Bolger & Schilling, 1991*) and dwell on negative concerns that might cause sleep disruptions, such as difficulty going to sleep (*Norlander, Johansson & Bood, 2005*). Positive affect strengthens the association between extraversion and improved sleep quality, whereas the tendency to experience positive affect mediates the relationship between neuroticism and sleep problems (*Allen, Magee & Vella, 2016*). People high in neuroticism and low in conscientiousness have poor sleep quality because these personality traits are associated with difficulty regulating emotions and behavior (*Duggan et al., 2014*).

## IMPLICATIONS

This study analyses the impact of smartphone usage on sleep quality while considering PT. Research has suggested that certain psychometrics could be predicted from the smartphone usage patterns, such as sensation seeking (*Mehrotra et al., 2017*; *Schoedel et al., 2018*). Social media data have successfully been utilized to predict PT (*Settanni, Azucar & Marengo, 2018*), and some studies have already started to consider smartphone usage data to achieve this goal (*Chittaranjan, Blom & Gatica-Perez, 2013*). In this context, our study is aligned with the growing body of literature investigating the relationship between smartphone usage and a range of psychometrics. It aids in extracting both dimensions of sleeping quality and smartphone usage patterns from daily smartphone usage (*Lee et al., 2021*). It could then be utilized to add features that predict and build on personality when designing interventions to aid more self-regulated smartphone usage and hence enhance sleep quality.

Primarily, our findings can be used to build smart social media and smartphone applications that are more accountable to users' digital well-being, where personalization is one of the acceptance factors (*Almourad et al., 2021*) and sleep quality is the main well-being factor. Similar to other domains such as drug abuse (*Conrod, 2016*), intervention and preventative methods can be personality-tailored also in the case of smartphone users. For instance, when a smartphone user is a poor sleeper who is also high in conscientiousness, digital well-being services, such as Google Digital Wellbeing (https://wellbeing.google/) can tailor their suggestions and recommendations to promote features related to mindfulness and self-regulation, such as limit setting. However, for poor sleepers who are low in conscientiousness, the same features shall be augmented with short-term incentives, regular feedback, and reminders to attain the same goal (*Cham et al., 2019*). Additionally, our findings showed that smartphone users who score low on conscientiousness and high on neuroticism may be recognized as having a considerably higher risk for poor sleep quality. The combination suggests proneness to high stress (*Carver & Connor-Smith, 2010*), and recommendations and features shall support stress coping strategies (*Barrick & Mount, 2015*), knowing that smartphone usage can be a way to escape stress; one of the symptoms of technology addiction (*Young, 1998*).

Smartphone users suffering from insomnia, and personality could help tailor interventions to improve adherence and, ultimately, sleep quality (*Al Battashi et al., 2021*). The chance is high that insomnia patients could use smartphones as a coping technique. Knowing their PT, perhaps through analysing the usage pattern of the smartphone itself, helps personalize established solutions to insomnia, such as cognitive behavioral therapy (*Dupuy et al., 2022*; *Yu et al., 2019*).

## CONCLUSIONS AND FUTURE DIRECTIONS

In this study, we tested the effects of objectively measured smartphone usage, PT, age, and gender on four sleep variables, *i.e.,* sleep duration, sleep distraction, sleeping time, and wakeup time. Through regression analysis, it was revealed that increasing smartphone usage predicted reduced sleep duration, sleep distraction, and later bedtime. This is consistent

with existing sleep-media literature (*Ayala et al., 2017*; *Costa, Patrão & Machado, 2019*; *Pensuksan et al., 2016*; *Yazdi et al., 2016*). However, smartphone usage was not associated with wake-up time. This study revealed that smartphone users with conscientiousness have a positive association with sleep duration, and early sleeping time, and a negative association with sleep distraction and poor wake-up time. Openness has a positive association with sleep distraction. Extraversion and neuroticism have been observed as positive contributors to the prediction of early walkability. Neuroticism has a negative association with early walkability.

The current findings add to the rising body of literature documenting the association between smartphone users' personalities and sleep behaviors. On the dark side, being too dependent on smartphones has consequent adverse effects on sleep duration, distraction, sleepability, and wakeability.

Our study has several limitations. We considered only smartphone usage, whereas the users can also be using other digital devices such as tablets and laptops. Hence, our results should be interpreted with caution and not generalized to technology use before bedtime. Also, the user's goal and intention of using particular applications before bed were not collected in this work. This could have added more context to our analysis and helped classify technology use based on the purpose. For example, YouTube can be used as a sleep aid, and passive viewing, such as watching videos with relaxing music and scenery, and the same app can be used in a more active style to watch live streaming and participate in commenting on it. Another limitation of our study relates to whether participants had flexible work schedules that might have allowed them to sleep at non-ordinary hours. We have classified sleeping time as early, regular, delayed, and poor based on common standards. Regression models can be improved by adding more variables related to participants' physical and mental health. For instance, studies have shown that anxious people have more unstable circadian rhythms, which can cause delayed sleep (*Kim, 2013*). Our results should hence be interpreted with caution and be better limited to smartphone use. This study can be extended while considering the diversity of apps being used *via* smartphones and PT in the prediction of sleep variables. The relation between PT, smartphone usage, and sleeping variables could be explored further while using models like the structured equation model. For example, a personality trait like neuroticism could be a moderator for the relation between the amount of time used before sleep and the parameters of sleep quality.

In future work, we plan to study the type of applications used before sleep and those used while the person sleep is distracted. This shall help a deeper understanding of the impact of different types of applications and their relation to sleep. We will also investigate the impact of applications used before sleep on sleep and the relation between PT and that near bedtime smartphone usage.

### Funding

The authors received no funding for this work.

## Competing Interests

The authors declare there are no competing interests.

## Author Contributions

- Aftab Alam conceived and designed the experiments, performed the experiments, analyzed the data, performed the computation work, prepared figures and/or tables, authored or reviewed drafts of the article, and approved the final draft.
- Sameha Alshakhsi conceived and designed the experiments, analyzed the data, authored or reviewed drafts of the article, and approved the final draft.
- Dena Al-Thani conceived and designed the experiments, authored or reviewed drafts of the article, and approved the final draft.
- Raian Ali conceived and designed the experiments, authored or reviewed drafts of the article, and approved the final draft.

## Ethics

The following information was supplied relating to ethical approvals (*i.e.*, approving body and any reference numbers):

The study has been approved by Hamad Bin Khalifa University Institutional Review Board (approval number 2021-08-102).

## Data Availability

The data and data dictionary are available in the Supplemental File.

## Supplemental Information

Supplemental information for this article can be found online at http://dx.doi.org/10.7717/peerj-cs.1261#supplemental-information.

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
