# Peer review of "The role of objectively recorded smartphone usage and personality traits in sleep quality"

_PeerJ Computer Science, doi:10.7717/peerj-cs.1261_

## Round 0.1 · original submission · Minor Revisions

The reviewer's comments must be addressed. Figures and section must be organised to a very good standard.

Reviewer 1 ·

Basic reporting

The objective of the present study is to assess the role of PT and smartphone usage in sleep quality. The paper is well presented, and rigorous experimentation has been carried out. I

1. The introduction of the paper is too long. Authors should consider moving the related works to a separate section. Though many research works have been reviewed, there is no effort of concluding the papers. Authors may be considering presenting the main points of each work in tabular form.
2. The language of the paper must be refined to improve readability, like “Line 138,139,10: The privacy policy of the app, to which all users agree while installing the app, 139 also states that data can be used for research. completed the 10-Item Big Five Inventory (BFI-10) 140 (Rammstedt and John, 2007) along with demographic information;” Line 190 From the smartphone usage data, we extract three continuous, i.e., e-sleep duration, e- 191 sleep distraction, and the daily average of smartphone usage, and four categorical variables, i.e., 192 e-sleep duration quality, e-sleep distraction category, e-sleeping time, e-wake-up time. How can something continuous like e-sleeping time, and e-wake-up time be considered categorical?
3. Authors need to specify how they assess the e-sleep duration quality and e-sleep distraction category.
4. There are some inconsistencies in how the study design is presented, as in line 132, and the author states In the 1st phase, the data were collected via a dedicated mobile app meant to allow users to know the time they spend on the smartphone and the apps used. And again in line 141 authors state “In the 2nd phase, the data were collected.”
5. The authors have cited Figure 1. in Line 132 and Figure 3 in line 145. Any special reason for not citing Figure 2?
6. Acronyms should be defined in their first occurrence like in Line 245 Sur.
7.

Experimental design

Through the experiments, the users have collected data about the subjects and used one or the other models to predict one or more characteristics of the subject. Is my concern, how the authors establish the ground truth to train the model in the first place.
8. The authors have stated in Line 242 that First, the assumptions were checked and verified. The 243 dependent variable SDur was visually checked for being normally distributed. Why didn’t they consider any statistical testing?

Validity of the findings

9. Authors should have discussed the possibility of external factors in their study, like occupation, the physical condition of the subjects

Reviewer 2 ·

Basic reporting

1. readability of the work is poor as no conceptual modeling and validation based explanation of individual concept design have been followed.
2. significance of the work is not clear in relation of the literature. Why part of the research is not clearly related to literature.

Experimental design

1. poor design as a very limited description of research methods without any modeling of the problem statement.

Validity of the findings

results have not been analyzed critically. It is discussed as what is being obtained in the research but how, and why part of the analysis is missing in the discussion.

Additional comments

Overall, below standard research without any relation to recent existing literature.
Very similar studies have been conducted before and they are not used as a comparative analysis of results and concepts presented in the paper.

Reviewer 3 ·

Basic reporting

1. Very good writing of the paper and structure is clearly readable and scientific standard.
2. references are up to date and recent.
3. results are discussed with clarity and detailed analysis.

Minor suggestion for readability:
1. In fig. 4, why you use different shapes for processing steps "e-sleep.." is not clear. Please clarify in the fig. description.
2. Results present in Tables should also be presented as graphs for better readability such as Table 4, and 6

Experimental design

1. clear experiment design and presentation
2. in-depth and analytical research discussion in experiment section

Validity of the findings

1.suficiently novel finding with clear benefits of the output presented.
2. can be very useful for literature progress and for readers in the particular domain of research

Additional comments

overall, very good quality of research.

---

## Round 0.2 · accepted · Accept

The paper has been edited and revised based on the reviewer's comments.

Reviewer 2 ·

Basic reporting

Very good paper structure. It is been improved good.

Experimental design

All comments have been addressed carefully.

Validity of the findings

visible contribution to findings.

Additional comments

very good research theme. all comments have been addressed.